# Root-Associated Microbiomes of *Panax notoginseng* under the Combined Effect of Plant Development and *Alpinia officinarum* Hance Essential Oil

**DOI:** 10.3390/molecules27186014

**Published:** 2022-09-15

**Authors:** Han-Lin Zhou, Xiao-Yun Liu, Chao Zhou, Si-Fang Han, Fu-Rong Xu, Xian Dong

**Affiliations:** 1School of Chinese Materia Medica, Yunnan University of Chinese Medicine, Kunming 650500, China; 2Yichang Key Laboratory of Omics-Based Breeding for Chinese Medicines, Key Laboratory of Three Gorges Regional Plant Genetics and Germplasm Enhancement (CTGU), Biotechnology Research Center, China Three Gorges University, Yichang 443002, China; 3Institute for Systems Biology, School of Life Sciences, Jianghan University, Wuhan 430056, China

**Keywords:** rhizosphere microbiomes, natural products, root rot disease, plant pesticide, saponins

## Abstract

Essential oils (EOs) have been proposed as an alternative to conventional pesticides to inhibit fungal pathogens. However, the application of EOs is considerably limited due to their highly volatile nature and unpredictable effects on other microbes. In our study, the composition of bacterial and fungal communities from the rhizosphere soil of *P. notoginseng* under four treatment levels of *Alpinia officinarum* Hance EO was characterized over several growth stages. Leaf weight varied dramatically among the four EO treatment levels after four months of growth, and the disease index at a low concentration (0.14 mg/g) of EO addition was the lowest among the *P. notoginseng* growth stages. The content of monomeric saponins was elevated when EO was added. Bacterial and fungal diversity in the absence of plants showed a decreasing trend with increasing levels of EO. Bacterial diversity recovery was more correlated with plant growth than was fungal diversity recovery. Compared with the control (no EO addition), a low concentration of EO significantly accumulated *Actinomycota*, including *Acidothermus*, *Blastococcus*, *Catenulispora*, *Conexibacter*, *Rhodococcus*, and *Sinomonas*, after one month of plant-microbial interaction. Overall, the results showed that both the plant growth stage and EOs drive changes in the microbial community composition in the rhizosphere of *P. notoginseng*. Plant development status had a stronger influence on bacterial diversity than on fungal diversity. EO had a more significant effect on fungal community composition, increasing the dominance of *Ascomycota* when EO concentration was increased. Under the interaction of *P. notoginseng* growth and EO, a large number of bacterial genera that have been described as plant growth-promoting rhizobacteria (PGPR) responded positively to low concentrations of EO application, suggesting that EO may recruit beneficial microbes in the root zone to cope with pathogens and reduce root rot disease. These results offer novel insights into the relationship between EO application, altered microbial communities in the plant roots, plant growth stage, and disease occurrence.

## 1. Introduction

*Panax notoginseng* (Burk.) F.H. Chen, belonging to the ginseng genus of the Araliaceae family, functions in blood circulation activation and pain alleviation and has high medicinal and economic value [1]. Continuous cropping obstacles can decrease the yield and quality of *P. notoginseng*, thus affecting its effective components and its medicinal efficacy [2]. The prevention and control of *P. notoginseng* disease frequently relies solely on the application of the chemical fungicides, such as hymexazol, dimethomorph, and metalaxyl, which have a large negative influence on the environment and human health [3]. With widespread public concern about the long-term environmental and health effects of conventional synthetic pesticides, natural products derived from plants have long been considered as potential alternatives [4]. Volatile chemical compounds extracted from plants, known as essential oils (EOs), have long been studied for their antimicrobial activity and are still popular in biomedical research nowadays [5]. The market request of EOs has been increasing with their wide range of applications in foods, pharmaceuticals, cosmetics, perfumes, aromatherapy, and agriculture. Some plant EOs not only repel insects but also have contact and fumigant insecticidal effects on specific pests and fungicidal effects on some important plant pathogens [6]. EOs may serve as alternative crop protectants and are attracting increasing interest for research as well as diverse applications. Perhaps the most attractive aspect of using EOs and/or their components as crop protectants (and in other contexts for pest management) is their low toxicity to mammals. Since many EOs and their components are commonly utilized as culinary herbs and spices, pesticide products containing certain EOs are exempt from the toxicity data requirements by the Environmental Protection Agency (EPA) [6].

EOs are a broad group of lipophilic compounds with low molecular weight and a high vapor pressure that facilitates evaporation and diffusion over long distances through porous soils. Large amounts of EOs for commercial use can be obtained via classical methods such as distillation, organic solvent extraction, and cold pressing. Currently, research is mainly focused on developing innovative and eco-friendly techniques to extract EOs and subsequently stabilize them through encapsulation to obtain GRAS (generally recognized as safe) natural products. Some previous studies mentioned that volatile compounds produced by plants have distinct bioactive functions, including promoting plant growth, activating plant resistance against pathogen infection, and inhibiting pathogens. Additionally, various studies have demonstrated that microbial volatiles can significantly reduce the viability and proliferation of devastating plant pathogens such as *Botrytis cinerea*, *Fusarium oxysporum,* or *Magnaporthe oryzae* [7,8]. In many cases, soil disease suppression is microbial in nature and is unrelated to the physical and chemical properties of the soil. EOs are low-molecular-mass metabolites that are involved in long-distance interactions with potent antimicrobial activity [9,10]. In addition, it has been shown that plant root exudates in soil can leave a long-lasting effect on the soil microbiome composition and pathogen control [11]. As such, it is clear that plant growth manipulation in the field are important strategies for soil-borne disease management. However, the role of EOs in the soil in the presence of both pathogens and host plants is yet unexplored. We anticipated that specific rhizobacterial and rhizosphere fungal taxa and volatiles would correlate with disease suppression. It is interesting to note that, while there is a wealth of information on the in vitro effects of volatiles on microbes, little is known about their effects on plant pathogens in vivo [12,13].

The development of plant EO-based pesticides from the laboratory to commercial use is expensive and time-consuming. Toxicological and environmental tests are normally required to register products [4]. Strikingly, very few studies have examined the chronic (long-term exposure) effects of EOs. Therefore, understanding how rhizosphere microbial communities respond to EO and plant physiology is of crucial importance. The toxicity and deterrence of plant EOs against plant-infecting pathogens have been well documented in the laboratory, but whether these will be interpreted as viability as crop protectants in vivo remains unclear. Plants are in constant chemical communication with organisms below ground through root exudates. We lack a comprehensive understanding of how *Panax notoginseng* grows under continuously cropped soil with different levels of added EO, particularly how the EOs may adjust the quantities and composition of bacterial and fungal communities in the rhizosphere soil. In this study, we analyzed rhizosphere samples across three growth stages of *Panax notoginseng* to gain insight into the interactions among *P. notoginseng*, EOs, and microbial communities. Hence, understanding how these interactions impact the soil microbiome composition, which opens up potential opportunities to explore beneficial outcomes of agricultural management associated with soil-borne disease control.

## 2. Results

### 2.1. Effects of EO on Plant Growth, Disease Occurrence and Saponin Content

There were varying degrees of increase in root, stem, and leaf weight and plant height under different EO treatments, especially for stem and leaf weight under the low concentration of EO (0.14 mg/g). A high concentration of EO (0.56 mg/g) had no effect on plant growth compared with no EO addition (Figure 1A–D).

As *P. notoginseng* grew in continuously cropped soil, the disease index began to increase. However, the disease indices of plants cultivated under different concentrations of EO were significantly decreased. At the fourth stage, the disease index of the low EO concentration treatment was 23.18, whereas the disease indices of the high EO concentration treatment and control were as high as 50.59 and 52.02, respectively (Figure 1E).

After a month of cultivation, the Ginsenoside Rd content in the EO treatments was greatly increased compared with that in the control; in particular, the most significant increase was observed in the low EO concentration treatment, from 11.4 to 33.6 mg/g. The Ginsenoside Rb1 content of the low EO concentration treatment increased from 9.2 to 20.1 mg/g compared to that of the control in the first month, and then a slight decrease was observed in the third month. In the other two treatments, there was no significant change in content compared with the control. The Ginsenoside Rg1 content was significantly increased after EO was added, but the increase became insignificant after three months compared with the control. The Notoginsenoside R1 content was not significantly changed compared with the control during the whole cultivation stage (Table 1). Compared with the control, the total saponin content increased significantly from 43.8 to 86.0, 70.2 and 85.3 mg/g in the low, medium, and high EO concentration treatments, respectively, but the increase gradually decreased after three months (Table 1).

A low concentration of EO can reduce the incidence index of root rot disease and promote the accumulation of biomass and saponin content.

### 2.2. Different Responses of Bacterial and Fungal Diversity to Plant Development and EO Treatment

A total of 6,839,032 and 8,027,502 (bacteria/fungi) raw reads were obtained in our datasets. Reads that could not be assembled were discarded, and 22,696 reads for bacteria and 3980 reads for fungi were filtered, respectively. The average read length after filtration was 300 bp for bacteria and 272.45 bp for fungi. The sequences retained in each sample were processed to generate 8901 and 3022 (bacteria/fungi) OTUs.

Before *P. notoginseng* was planted, the soil was treated with different concentrations of EO for seven days. Low EO concentrations were found to improve the bacterial diversity, while the medium and high EO concentration treatments reduced the bacterial diversity, which was more obvious in the high EO concentration treatment. After a month of *P. notoginseng* growth in the soil, the bacterial diversity was restored to different degrees, and the soil without EO had the most abundant bacterial diversity. The low EO concentration treatment had a relatively high bacterial diversity, followed by the medium EO concentration treatment, and the high EO concentration treatment had the lowest diversity. After planting *P. notoginseng* for three months, the bacterial diversity did not change significantly under the control and low EO concentration treatments, while the bacterial diversity under medium and high EO concentration treatments increased significantly compared with the two prior months (Figure 2A).

With increasing EO concentration, the fungal diversity decreased after treatment with different EO concentrations before *P. notoginseng* growth. After *P. notoginseng* was planted, under the combined effects of the EO and *P. notoginseng*, the fungal diversity in the treatments continued to decrease except for in the low EO concentration treatment, which increase the diversity of fungi. The high EO concentration treatment had the greatest reduction in fungal diversity. After planting *P. notoginseng* for three months, compared with the two prior months, the diversity of fungi showed different degrees of recovery. Compared with the period when *P. notoginseng* was not planted, the soil fungal diversity was decreased in the treatments without EO, while the soil fungal diversity was increased by adding EO, especially after three months at medium concentrations (Figure 2B).

In total, the EO had no significant effect on bacterial and fungal diversity except in the first month of plant growth at a high EO concentration, which significantly reduced fungal diversity. Bacterial diversity recovery was more correlated with plant growth than fungal diversity recovery was.

### 2.3. Bacterial and Fungal Community Composition in Response to Plant Advancement and EO Treatment

High-throughput sequencing of the rhizosphere samples at all three development stages and four EO levels was performed. The bacterial community compositions in the rhizosphere under different growth stages and EO levels are shown in Figure 3A. Before *P. notoginseng* was planted, the relative proportions of *Proteobacteria*, *Actinobacteria*, *Chloroflexi*, *Acidobacteria,* and *Firmicutes* were dominant, among which the proportion of *Actinobacteria* was the highest, followed by *Proteobacteria* and *Chloroflexi* and then *Acidobacteria* and *Firmicutes*. Principal coordinate analysis (PCoA) was performed at OTU level. The rhizosphere bacterial community structure was clustered during the pretreatment stage but clearly separated during the plant growth stage (Figure 3C). In the absence of plant growth, EO decreased the proportion of Bacteroidetes and increased the proportion of *Firmicutes*. The high concentration of EO reduced the proportion of *Bacteroidetes*, and the reduction rate was approximately 50% compared with the no EO treatment. The proportion of *Firmicutes* in the high-concentration EO treatment was significantly higher than that in the lower-concentration EO treatment. *Firmicutes* accounted for 3.8% in the soil treated with a high EO concentration, 2.8% in the medium EO concentration soil, and approximately 1.6% in the low EO concentration soil (Figure 4A and Appendix A). After a month of interaction between EO and plant growth, the proportion of *Actinobacteria* was significantly increased in the soil treated with both high and medium concentrations of EO compared with that in the soil treated with EO alone, and the proportion of *Actinobacteria* in the soil treated with a high concentration of EO was 31.5%. After three months of interaction between EO and plant growth, the proportion of *Acidobacteria* in the soil treated with a low concentration of EO was decreased compared with the control, and the proportion of *Actinobacteria* was increased to 26.1%. *Actinobacteria* was increased to 23.5% in the soil treated with a medium concentration of EO (Figure 4A and Appendix A). Therefore, plants had a greater influence on bacterial community composition than EOs.

The fungal community compositions in the rhizosphere samples under different growth stages and EO levels are shown in Figure 3B. Rhizosphere fungal communities were affected by EO levels and plant growth stages. At the pretreatment stage, the rhizosphere fungal community structure was obviously differentiated by EO level. Under the joint effects of plants and EO levels, the difference in the fungal community between different EO levels increased (Figure 3D and Figure 4B). Compared with the control, EO at different concentrations expanded the proportion of *Ascomycota* and decreased the proportion of *Basidiomycota* to different degrees. The proportion of *Ascomycota* was increased to 73.9%, 61.2%, and 51% in the high, medium, and low EO concentration treatments, respectively. With increasing EO concentration, the proportion of *Basidiomycota* decreased to 30.7%, 20.8%, and 8%, respectively (Figure 3B and Appendix A). After a month of *P. notoginseng* growth, the proportion of *Ascomycota* decreased, the proportion of *Basidiomycota* increased significantly, and the proportion of *Mortierellomycota* also increased slightly. Low and medium EO concentration treatments showed similar change trends, but the change range was not as significant as that of the blank control. Under the high-concentration EO treatment, except for the decrease in *Basidiomycota* proportion, the change trends of other species were consistent with those of other EO treatments. After three months of *P. notoginseng* growth, the composition of the soil fungal community was similar between the low and medium EO concentrations, and the proportion of *Ascomycota* was higher than that of the control. Compared with one month after *P. notoginseng* planting, the proportion of *Basidiomycota* was significantly increased and the proportion of *Mortierellomycota* was significantly decreased under the high EO concentration treatment. Therefore, the effect of EO on the composition of the fungal community was stronger than that of plants, but this effect gradually weakened with the decrease in EO content in the soil (Figure 3B,D and Figure 4B).

### 2.4. Correlations of Plant Biomass, Disease Index, and Saponin Content with Bacteria and Fungi

The Mantel test showed that leaf weight correlated significantly with *Acidobacteria*, *Actinobacteria*, *BRC1*, *Elusimicrobia,* and *Planctomycetes*. Root weight correlated significantly with *Chlamydiae*, *Dependentiae*, *Elusimicrobia,* and *Nitrospirae*. The occurrence of disease was correlated with the bacteria mentioned above and with *Gemmatimonadetes* (Figure 5A). For fungal community composition, leaf weight, root weight, and disease occurrence positively correlated with *Glomeromycota* and *Mortierellomycota*. Plant height was correlated with *Chytridiomycota*. Leaf weight was also correlated with *Mucoromycota* (Figure 5B).

To get an in-depth understanding of interactions within bacterial communities, relationships between bacteria were measured. *Acidobacteria* was significantly positively correlated with *Latescibacteria*, *Elusimicrobia*, *BRC1,* and *Armatimonadetes* and negatively correlated with *Actinobacteria*. *Actinobacteria* negatively correlated with *Nitrospirae*, *Firmicutes*, *Elusimicrobia*, *BRC1,* and *Armatimonadetes*. *Armatimonadetes* positively correlated with *Verrucomicrobia*, *Elusimicrobia,* and *BRC1*. *Bacteroidetes* positively correlated with *Proteobacteria* and negatively correlated with *Chloroflexi*. *Chlamydiae* positively correlated with *Nitrospirae*, *Latescibacteria*, *Gemmatimonadetes*, *Elusimicrobia,* and *Dependentiae*. *Chloroflexi* positively correlated with *Planctomycetes* (Figure 5A). For fungal community interactions, *Cercozoa* was positively correlated with *Rozellomycota*, *Mucoromycota,* and *Glomeromycota* (Figure 5B).

Several bacteria were correlated with saponin content (Appendix A). Rb1, Rd, and Rg1 were significantly correlated with *Firmicutes* and *Latescibacteria*. Rd was correlated with *Actinobacteria*, *Chlamydiae*, *Dependentiae,* and *Elusimicrobia*. *Dependentiae* significantly correlated with Rg1. The Rd content was correlated with *Mortierellomycota* (Appendix A).

## 3. Materials and Methods

### 3.1. General Experimental Design

The EO, which had antifungal activity against pathogenic fungi causing root rot of P. notoginseng, was added to the continuous cropping soil for seven days, and the changes of rhizosphere microbiome were mainly caused by different concentrations of the EO. After planting *P. notoginseng* into treated soil, the relationship between the change of microbiome and the occurrence of *P. notoginseng* root rot was studied under the joint action of plants and EO. The entire experiment was carried out in glasshouse with three replicates per treatment. Each replicate contained 10 individual pots. Five *P. notoginseng* seedling was transplanted in each pot.

### 3.2. Preparation of EOs

*Alpinia officinarum* Hance was purchased from the Yunnan Jinfa Pharmaceutical Limited Company (Kunming, China) and authenticated by Prof. Bin Qiu at Yunnan University of Chinese medicine. EO was prepared from the dry rhizomes of *A. officinarum* by steam distillation for 7 h with 8-fold water (*v*/*w*). The EO was collected and dried by sodium sulfate and after that stored at −20 °C before use.

### 3.3. Pot Experiment and Sample Collection

Different amounts of EOs (0, 0.14, 0.28, and 0.56 mg g^−1^ soil) were added to continuously cropped soil to obtain four EO levels. Continuously cropped soil was obtained after cultivating *P. notoginseng* in it for three consecutive years. After mechanical stirring to mix it evenly, we closed the soil and kept it in a cool, dull put for 5 days. After the fumigation process, 450 g of that soil was moved into each pot. Five yearly seedlings of *P. notoginseng* were planted in each pot. Each group contained 150 plants. Soil collected after fumigation was referred to as pretreatment. Rhizosphere soil was collected after *P. notoginseng* growth for 1 month and 3 months. Rhizosphere soils from five plants were mixed together as one soil sample at each development stage, and each treatment contained five replicates. The rhizosphere samples in this study were entirely characterized as the soil within 2 mm of the root surface [14]. After gently shaking the roots to evacuate freely connected soil clumps, the rhizosphere samples were carefully collected by brushing the remaining soil off the roots [15].

### 3.4. Determination of Biomass, Disease Index, and Saponin Content

The biomass and disease index of *P. notoginseng* in each gather were measured each month during the cultivation period. The saponin content was measured after 1 month and 3 months. Four months after planting, the plants were graded for severity of disease using the values 0 (not showing chlorosis), 1 (leaf wilting), 2 (both leaf and stem wilting), or 3 (whole plant wilting). Each group contained 150 *P. notoginseng* plants. The disease index was calculated as follows: Disease index = Σ(diseased of grade × number of plants grade)/(total plants × the highest grade) × 100.

All the samples were dried and crushed, and 0.1 g of the powdered sample was weighed and mixed with 3.0 mL of pure methanol. The mixture was sealed, ultrasonically extracted at room temperature for 1 h (40 kHz, 300 W), and centrifuged at 3000 rpm for 10 min. The upper layer was collected, filtered through a 0.22-mm microporous membrane, and transferred to a sample vial. The vial was injected into a column for HPLC analysis to measure the R1, Rg1, Rb1, and Rd contents of *P. notoginseng*. The chromatographic column was an Agilent Zorbax Eclipse Plus C18 (4.6 mm × 250 mm, 5 μm). The mobile phase comprised of acetonitrile (A) and water (B), and the elution procedure was as follows: 0 min, 19% A; 12 min, 19% A; 60 min, 36% A; 77 min, 36% A, then returned to the initial concentration and maintained for 10 min. The flow rate was 0.6 mL·min^−1^, and the injection volume was 10 μL. The column temperature was 30 °C, and the detection wavelength was 203 nm.

### 3.5. DNA Extraction and Amplicon Sequencing

Total genomic DNA was extracted from 0.5 g of rhizosphere soil by grinding the soil with liquid nitrogen utilizing an E.Z.N.A.^®^ Soil DNA Kit (Omega Biotek, Inc., Norcross, GA, USA) following the manufacturer’s protocol. DNA concentration and size were monitored on Nanodrop spectrophotometer (Thermo Fisher Scientifc, Waltham, MA, USA) and by 1% agarose gel electrophoresis and kept at −80 °C prior to further use. The diversity and community structure of bacteria and fungi were assessed using high-throughput sequencing investigation of the 16S rRNA gene within the V3–V4 hypervariable region and the 18S rRNA gene within the ITS regions. The 16S and 18S rRNA genes were amplified with the primer pairs 338F (5′-ACTCCTACGGGAGGCAGCAG-3′): 806R (5′-GGACTACHVGGGTWTCTAAT-3′) and ITS1F (TCCGTAGGTGAACCTGCGG): ITS2R (5-GACGCTTCTCCAGACTACAAT-3), respectively. The primers contained overhanging bases to connect the Illumina sequencing adapters and dual-index barcodes in a second round of PCR. The PCR products were examined by agarose gel electrophoresis and then purified using an AxyPrepDNA Gel Recovery Kit (AXYGEN). Subsequent eight-cycle PCR was carried out, adding dual-index barcodes and Illumina sequencing adapters to each sample, after which the PCR products were purified using the TruSeqTM DNA Sample Prep Kit. Equal molar amounts of the PCR products from each sample were mixed and sequenced utilizing the Illumina MiSeq PE300 platform (Illumina, San Diego, CA, USA) according to the standard protocols by Sinotech Genome Technology Co. (Shanghai, China). The raw sequencing reads were deposited into the NCBI Sequence Read Archive (SRA) database (Accession Numbers: PRJNA853255 and PRJNA853175).

### 3.6. Analysis of Sequencing Data

There were four treatments for each growth stage, and five repetitions were collected for each treatment. Raw sequences were prepared utilizing the Quantitative Insights Into Microbial Ecology pipeline (QIIME) (qiime2-2020.11) [16]. The adaptor sequence, barcode, and 30 low-quality bases at the end of each read were removed, after which forward and reverse reads were joined using the fastq-join [17] method with a minimum overlap of 10 bp and a maximum mismatch within the overlap region of 10%. Reads that could not be assembled were discarded. The number of sequences per sample was normalized by the size of the sequences and the sampling coverage. Then the optimized sequences were clustered into operational taxonomic units (OTUs) using UPARSE 7.0 with 97% sequence similarity level [18]. The most abundant sequence for each OTU was selected as a representative sequence. The SILVA 16S and 18S rRNA database were used as a reference database for bacteria and fungi, respectively.

## 4. Discussion

Risks associated with the utilization of synthetic pesticides have led to the growth of an environmental movement seeking sustainable alternatives in disease control. EOs are a major category of botanicals that began to be researched in the 1980s [19]. As important alternatives to synthetic fungicides, EOs have received increasing attention due to their great potential for reducing disease occurrence and for their volatile properties, and might serve as promising sources of eco-friendly, natural pesticides with less chemical resistance by pests.

Soil contains a variety of other microorganisms in addition to disease-causing microorganisms. In addition to the known pathogen *F. oxysporum*, the effects of EO on the regulation of other microorganisms and plant growth are unknown. Another problem that remains to be solved is the availability and persistence of EO in soil. Due to the essential function of bacteria and fungi in plant growth and disease occurrence, a comprehensive study of the responses of root-associated microbiomes to EO levels and plant growth is necessary. In our study, in continuously cropped soil, a low concentration of EO increased the biomass accumulation of *P. notoginseng* and reduced the occurrence of root rot disease (Figure 1A–E). Meanwhile, a low concentration of EO could also increase the content of saponins (Table 1). Various EOs are effective in controlling plant disease, mainly due to their direct antifungal activity and indirect induction of disease resistance [20,21]. The cell membrane is primarily composed of lipids (mainly phospholipids), proteins, and sugars and is considered the first barrier that separates a cell from the external environment to ensure the relative stability of the internal environment [22]. Strong antimicrobial activity was observed for a variety of EOs on both *Escherichia coli* and *Staphylococcus aureus* mediated by cell membrane damage and leakage of cytoplasmic components [23]. Previous studies have shown that EO can induce the defense responses of plants by enhancing POD, PAL, and GLU activity [24]. *Foeniculum vulgare* Mill. EO as a promising antifungal agent against *Fusarium solani* not only to control root rot disease but also to enhance plant growth and activate plant defense [25]. EO reduces postharvest anthracnose of *Carica papaya* L. and modulates defense-related gene expression linked to plant defense, pathogenesis-related protein, cell wall-degrading enzymes, oxidative stress, abiotic stress, and the phenylpropanoid pathway [26]. The saponin content also changed across EO levels at different *P. notoginseng* growth stages. A straightforward explanation is that the elevated EO levels changed the physiological status of the plants. On the one hand, the proportion of pathogens decreased due to EO treatment; on the other hand, EO may improve the disease resistance of *P. notoginseng* as a signal substance to reduce the number of pathogens.

In addition to soluble secondary compounds, plants release various volatile organic compounds (VOCs) involved in interactions with surrounding soil organisms [27]. VOCs are small compounds of low molecular weight and lipophilic character with high vapor pressure and low boiling points [28,29]. Due to their physico-chemical properties, VOCs can easily diffuse through gas- and water-filled pores and can, therefore, have a wide effective range in soil. EO is a secondary metabolite of plants and belongs to the same category as the VOCs released by roots. The bacterial community structures in the rhizosphere were clustered at the pretreatment arrangement but clearly separated during the plant growth stage. In this study, the EO level had no significant effects on the bacterial community structure (Figure 3A and Figure 4A). After *P. notoginseng* growth, the rhizosphere consisted of a diverse microbial community due to the large amounts of organic nutrients partially released by the plant roots called root exudates [30]. Since root exudates could regulate the microbial communities and are strongly related to root-released organic carbon [31], the increased differences among bacterial communities across EO levels might have been because the influence of the root exudates overrode the effect of EO in the rhizosphere. Plant growth-promoting rhizobacteria (PGPR) are in close contact with roots and can enhance the adaptive capacity of host plants in their environments [32]. In this study, the relative abundances of *Pseudarthrobacter*, *Massilia*, *Paenarthrobacter*, *Pseudomonas*, *Bacillus*, and *Enterobacteriaceae* in the rhizosphere were higher at the pretreatment phase (Appendix A), and these genera have been portrayed as important PGPR. After three months of growth, the relative abundances of *Pseudarthrobacter*, *Paenarthrobacter*, and *Enterobacteriaceae* under the low EO concentration treatment were higher than those in nontreated soil (Appendix A). Compared with no EO addition, a low concentration of EO significantly recruited *Actinomycota*, including *Acidothermus*, *Blastococcus*, *Catenulispora*, *Conexibacter*, *Rhodococcus*, and *Sinomonas*, after one month of plant-microbial interaction (Appendix A). One explanation for these results is that the EOs could promote the growth of some beneficial bacterial growth and that plant growth could interact with these EOs to change the bacterial community composition. Chemical pesticide application changed the microflora and killed the PGPR, while the EO did not interfere with the growth of PGPR.

Fungal communities within the rhizosphere were affected by the EO level and plant growth stage (Figure 3B and Figure 4B). The fungal community structures within the rhizosphere were clearly different across different EO levels at the pretreatment stage, which might have been because of the combined effect of root exudates and EO in the rhizosphere. EO treatment can increase the proportion of *Penicillium* and *Trichoderma* (Appendix A), which are used for biocontrol against soil-borne diseases. *Ascomycota* and *Basidiomycota* were the most affected by EO. With or without the influence of plant growth, EOs increased the extent of *Ascomycota* and decreased the proportion of *Basidiomycota* (Figure 3B, Figure 4B, and Appendix A).

In the USA and Chile, pesticides based on plant EOs can be used as a single agent, or in combination (tank-mixed) with other crop protectants to upgrade their viability in field trials [4,33]. Our previous studies also showed that various kinds of EOs mixed with chemical pesticides such as hymexazol could have great synergistic or additive effects and no antagonistic effect [34,35]. The good compatibility of EOs and chemical pesticides can potentially lower the overall quantities applied. Meanwhile, EOs can be combined with existing chemical drugs with good effects and a low toxicity to possibly mitigate or delay the development of resistance in pathogen populations, thus extending the service life of chemical pesticides.

From our point of view, the volatile nature of EOs is both an advantage and a disadvantage. An advantage of utilizing EO-based pesticides is the lack of harvest restrictions of treated crops, since EOs have a low toxicity to mammals and a limited environmental persistence. Disadvantages of these pesticides include their lack of persistence when used as stand-alone products, and two or more carefully timed applications may be required for satisfactory management of disease. Particularly in soil, EOs or their major compounds could be used as raw materials for microbial growth and can decompose into other substances. Eugenol and other EO compounds are also nonpersistent in soils under aerobic conditions. The half-life for α-terpineol ranges from 30–40 h, with complete degradation by 50 h. Eugenol is totally broken down into common organic acids by soil-borne *Pseudomonas* bacteria [36]. In our study, with the volatilization and degradation of EO in soil, there was a significant change in the composition of the microbial community during the initial stage caused by medium and high concentrations of EO but it gradually recovered due to the short persistence of EO (Figure 2). The short residual half-lives of EOs on plants also enhance their compatibility with biological control agents and reduce risks to nontarget microorganisms. Note, however, that the disadvantages of limited persistence and phytotoxicity could be mitigated by microencapsulation of EOs during formulation. As public concern increases for many conventional pesticide products, EO-based pesticides may become an increasingly popular choice for disease management [4].

Industrial exploitation of EOs must establish a number of parameters for good agricultural practices for the plant cultivation (e.g., genotypes, selection and orientation of plots and practices, harvest time, conditions and technical parameters, and extraction) to minimize the heterogeneity of EOs [37]. A number of compounds are commercially available at reasonable purity levels (95%), and EO producers and suppliers can frequently provide chemical specifications for even the most complex oils [6]. However, only a few reports have demonstrated the direct, in vivo antimicrobial activity of plant volatiles during microbial host infection and colonization. More investigation is necessary to define the stage of the infection process at which the volatiles act, either prior to or after the host colonization occurred. Additionally, since plant volatiles are frequently emitted as complex blend, it is important to determine if individual compounds are active or whether the mixture has additive or synergistic effects, probably due to the different action modes of the components in the mixture. The lack of research on volatile–pathogen interactions demonstrated that there may be a rich variety of plant–microbe interactions that are mediated by volatiles. More information on the different roles of volatiles in plant–pathogen interactions may give new, sustainable disease management alternatives for agriculture and facilitate the discovery of novel direct and indirect defense mechanisms against economically important plant diseases [38].

## 5. Conclusions

Both plant development and EO levels strongly affected the structure of root-associated microbiomes. The bacterial community structure was closely related to plant development, whereas the fungal community was correlated with the EO level. A large number of PGPR were enhanced under the low EO level, suggesting that EO could recruit beneficial microorganisms, which might be an important strategy for plants to cope with pathogen infection. Linalool, caryophyllene, decyl ester, and 1-decanol were abundant compounds of the *A. officinarum* EO. Their impact on specific pathogens and persistence in the soil environment is also an issue that needs further study in the later period. This study is a step toward further understanding how changes in microbial community composition mediate and reflect the effects of EO application in intensive agricultural ecosystems.

## Figures and Tables

**Figure 1 molecules-27-06014-f001:**
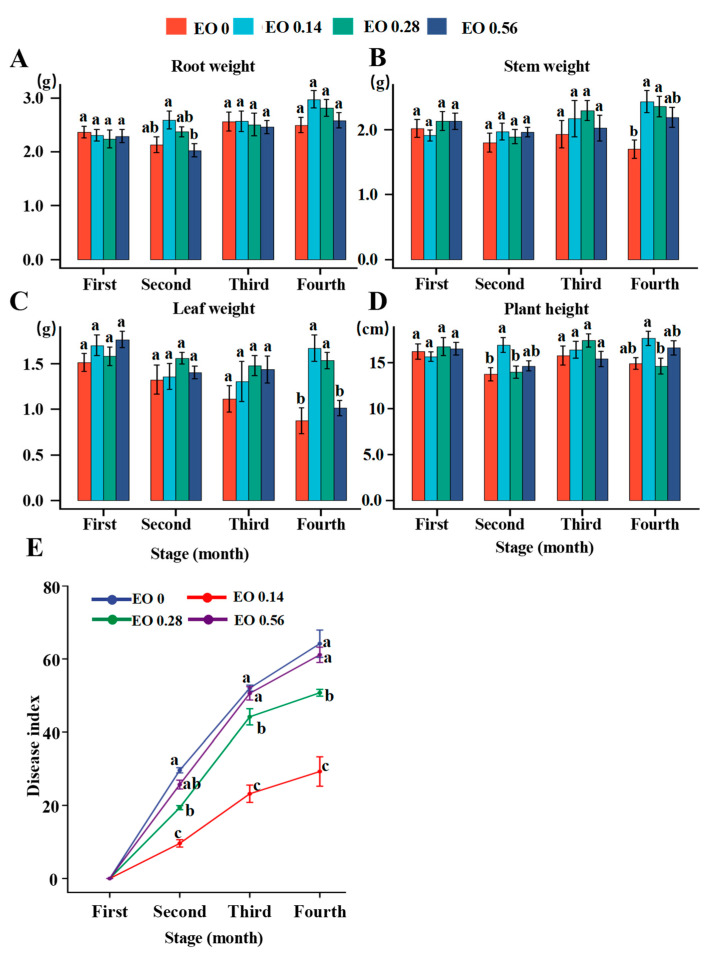
Root weight (**A**), stem weight (**B**), leaf weight (**C**), and plant height (**D**) in the four different essential oil (EO) levels at four stages. Error bars indicate the standard deviation of eight biological replicates. Different letters indicate significant differences (*p* < 0.05) between the EO levels at each growth stage. (**E**) Disease index of *P. notoginseng* among the four growth stages. The four stages were EO treatment for one month, two months, three months, and four months. The concentrations of EO in the soil were 0 mg/g (control), 0.14 mg/g (low concentration), 0.28 mg/g (medium concentration), and 0.56 mg/g (high concentration). Different letters in each stage indicate statistically significant differences (*p* < 0.05) according to ANOVA test.

**Figure 2 molecules-27-06014-f002:**
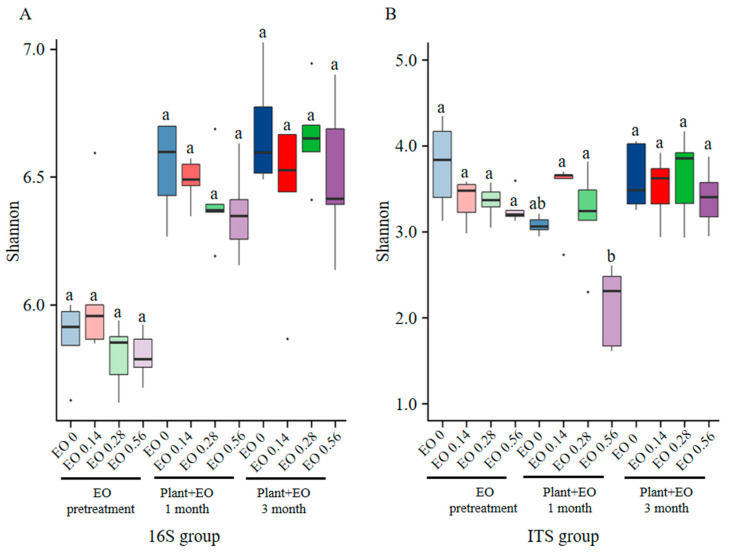
Box plots showing alpha diversity (Shannon index) variation across samples on 16S data (**A**) and ITS data (**B**). The concentrations of EO in the soil were 0 mg/g (control), 0.14 mg/g (low concentration), 0.28 mg/g (medium concentration), and 0.56 mg/g (high concentration). The three stages were: EO pretreatment for seven days, then *P. notoginseng* was planted and samples were assessed at one and three months. Different letters in each stage indicate statistically significant differences (*p* < 0.05) according to ANOVA test. Dots represent samples that deviate from other biological replicates.

**Figure 3 molecules-27-06014-f003:**
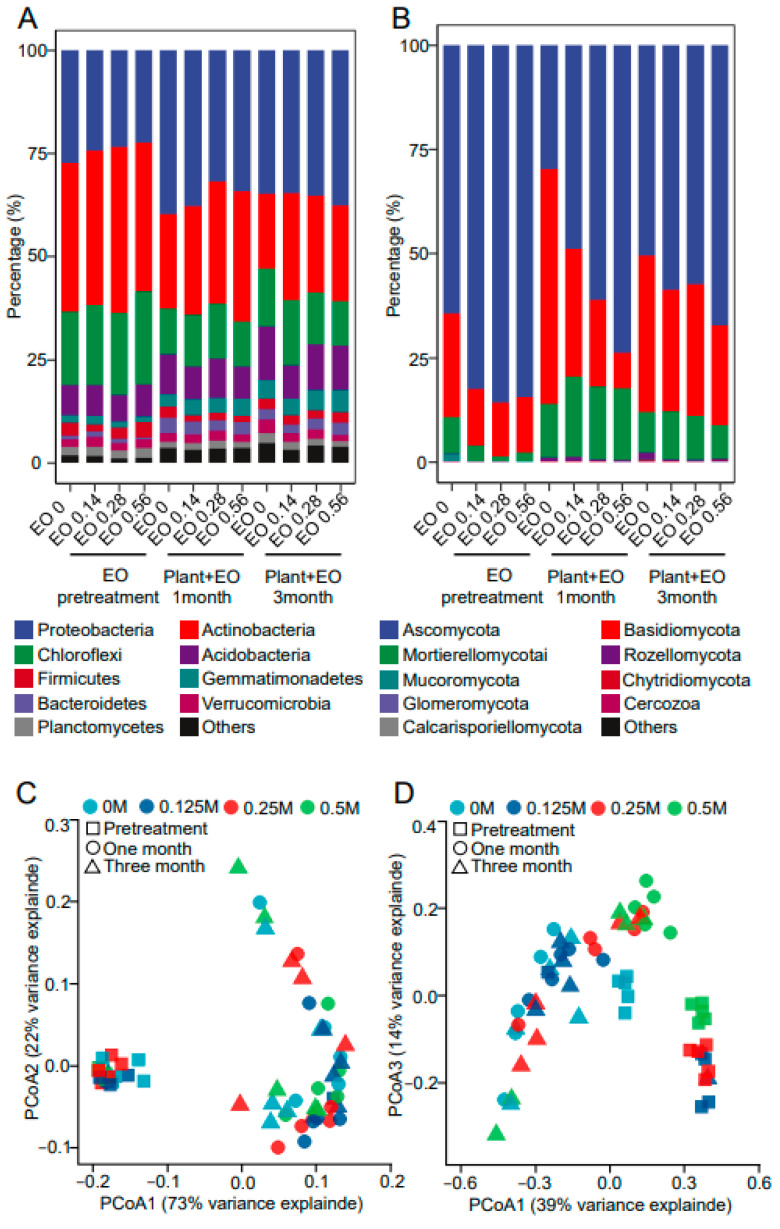
Bacterial (**A**) and fungal (**B**) community composition of the rhizosphere soil samples at the phylum level. Principal coordinate analysis (PCoA) of the microorganism communities in the 16S (**C**) and ITS (**D**) samples. PCoA and dissimilarity distance were based on the Bray–Curtis distance at the OTU level. The concentrations of EO in the soil were 0 mg/g (control), 0.14 mg/g (low concentration), 0.28 mg/g (medium concentration), and 0.56 mg/g (high concentration). The three stages were: EO pretreatment for seven days, then, *P. notoginseng* was planted and samples were assessed at one and three months.

**Figure 4 molecules-27-06014-f004:**
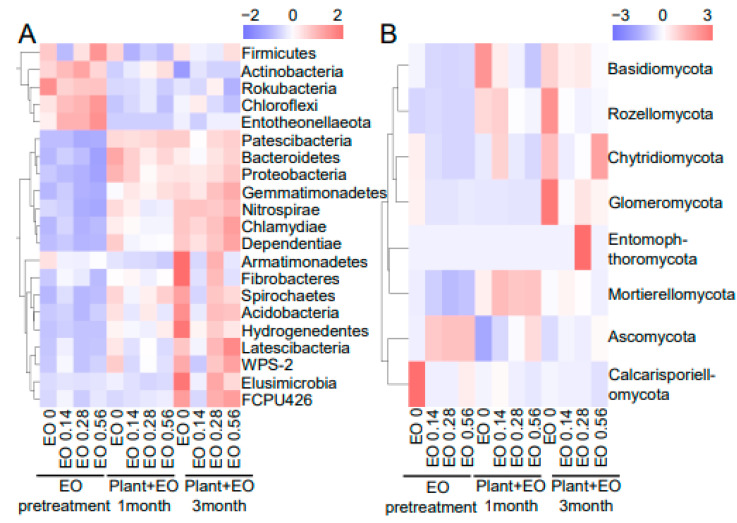
Heatmap of the dominant bacterial (**A**) and fungal (**B**) composition of the rhizosphere soil samples at the phylum level. The concentrations of EO in the soil were 0 mg/g (control), 0.14 mg/g (low concentration), 0.28 mg/g (medium concentration), and 0.56 mg/g (high concentration). The three stages were: EO pretreatment for seven days, then, *P. notoginseng* was planted and samples were assessed at one and three months.

**Figure 5 molecules-27-06014-f005:**
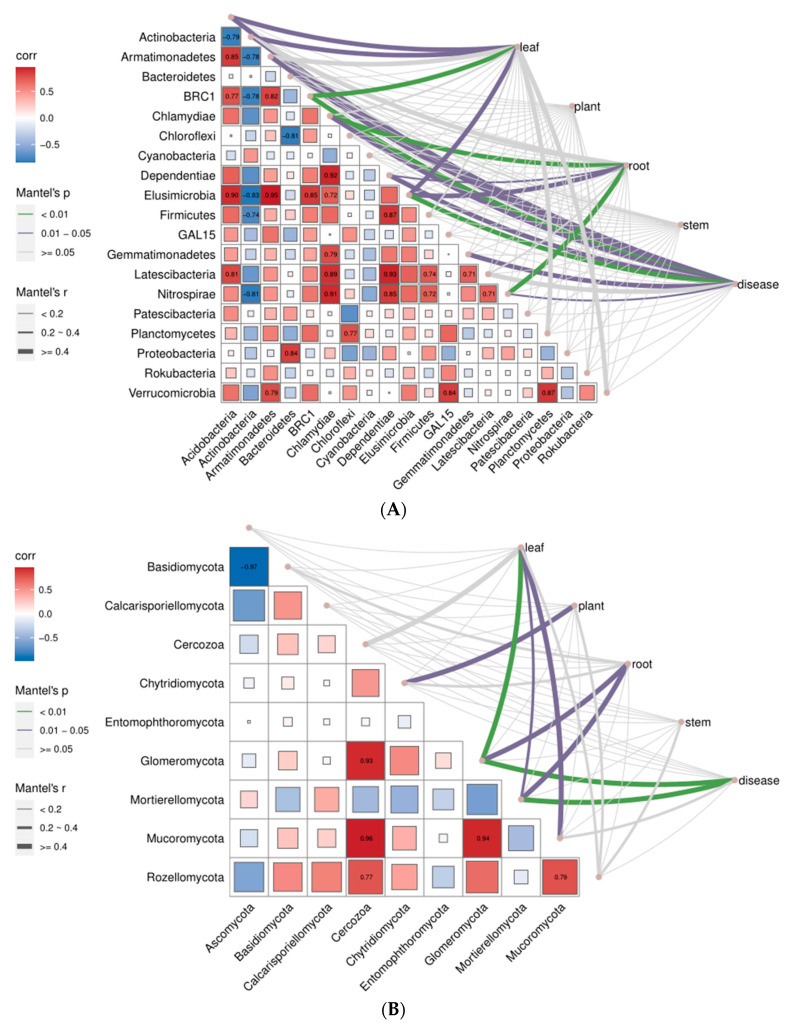
Pairwise comparisons of the phylum-level distribution of the 16S (**A**) and ITS (**B**) root microbiota are shown, with a color gradient that denotes Pearson’s correlation coefficients. Biomass (leaf weight, plant height, root weight, and stem weight) and disease index were correlated with the species of the community using the Mantel test. Edge width corresponds to Mantel’s r statistic for the corresponding distance correlations, and edge color denotes the statistical significance. The concentrations of EO in the soil were 0 mg/g (control), 0.14 mg/g (low concentration), 0.28 mg/g (medium concentration), and 0.56 mg/g (high concentration). The three stages were: EO pretreatment for seven days, then, *P. notoginseng* was planted and samples were assessed at one and three months.

**Table 1 molecules-27-06014-t001:** Variation of Notoginsenoside R1 (R1), Ginsenoside Rb1 (Rb1), Ginsenoside Rd (Rd), and Ginsenoside Rg1 (Rg1) content under different EO treatments. The concentrations of EO in the soil were 0 mg/g (control), 0.14 mg/g (low concentration), 0.28 mg/g (medium concentration), and 0.56 mg/g (high concentration). Different letters in the same column indicate significant differences (*p* < 0.05) between the EO levels at each growth stage.

Treatment	One Month				Three Months			
	R1	Rb1	Rd	Rg1	R1	Rb1	Rd	Rg1
EO 0	2.61 ± 0.25 ^b^	10.06 ± 1.68 ^c^	10.82 ± 1.03 ^b^	17.79 ± 2.12 ^b^	3.81 ± 0.06 ^a^	10.43 ± 0.61 ^b^	8.66 ± 1.13 ^b^	23.54 ± 4.5 ^a^
EO 0.14	3.70 ± 0.16 ^a^	20.28 ± 1.55 ^a^	34.39 ± 12.9 ^a^	23.45 ± 1.39 ^ab^	7.15 ± 3.44 ^a^	15.73 ± 0.51 ^a^	17.04 ± 1.63 ^a^	24.87 ± 1.44 ^a^
EO 0.28	2.45 ± 0.31 ^b^	14.93 ± 1.10 ^b^	20.79 ± 7.96 ^ab^	22.98 ± 1.58 ^ab^	4.55 ± 0.80 ^a^	10.07 ± 1.09 ^b^	7.70 ± 0.87 ^b^	20.30 ± 1.24 ^a^
EO 0.56	3.05 ± 0.64 ^ab^	17.59 ± 0.64 ^b^	26.81 ± 1.89 ^ab^	25.94 ± 3.49 ^a^	2.79 ± 0.36 ^a^	11.61 ± 0.9 ^b^	8.64 ± 2.66 ^b^	23.36 ± 1.83 ^a^

## Data Availability

All raw sequence data have been made available in the NCBI Sequence Read and Archive (SRA) database under the accession number PRJNA853255 and PRJNA853175, respectively.

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
