# Peer review of "Root-Associated Microbiomes of Panax notoginseng under the Combined Effect of Plant Development and Alpinia officinarum Hance Essential Oil"

_molecules, 2022, doi:10.3390/molecules27186014_

Round 1
Reviewer 1 Report
The work is well prepared, the topic is very interesting and topical. In my opinion, the article needs some minor corrections (below). I would like to add that a valuable element of this type of work is the analysis of the chemical composition of the oil used as a natural pesticide, especially that the authors cite works on specific compounds of the oil, their impact on specific pathogens and persistence in the soil environment. If the authors have such data, it would be good to include them, if not taken into account in further research, indicating it at the end of their conclusions.
1. Plant names should be written in italics
2. The abstract is too long, only the most important information on methods, results and conclusions should be left
3. It would be good to replace the keywords with other words than those used in the title (we will then get more phrases in search engines)
4. Fig. 1 Description needs improvement, no title for this drawing, incomprehensible descriptions on the horizontal axis (months? Stage?)
5. Line 125. Lack of information what the abbreviations Rd, Rb etc. mean. Similarly in table 1.
6. Analysis of the oil used?
Author Response
Reviewer 1:
The work is well prepared, the topic is very interesting and topical. In my opinion, the article needs some minor corrections (below). I would like to add that a valuable element of this type of work is the analysis of the chemical composition of the oil used as a natural pesticide, especially that the authors cite works on specific compounds of the oil, their impact on specific pathogens and persistence in the soil environment. If the authors have such data, it would be good to include them, if not taken into account in further research, indicating it at the end of their conclusions.
Response: Thank you very much for your positive comments. According to your valuable suggestions, we have carefully revised the full text and marked it with a yellow background. The chemical composition of EOs was analyzed by GC/MS. 85 compounds were identified from A. officinarum EO. Linalool (20.25%) was the most abundant component. The other abundant compounds were found to be caryophyllene (12.80%), decyl ester (7.03%), and 1-decanol (5.02%), respectively. Please check methods and results sections of our published paper [Sun, et al. Molecules, 2018, 23(5):1021]. We also indicated the impact of specific compounds on specific pathogens and persistence in the soil environment at the end of conclusions according to your suggestion.
- Plant names should be written in italics
Response: We have corrected them. Please check the yellow shading in the revised version.
- The abstract is too long, only the most important information on methods, results and conclusions should be left
Response: We have reduced the content of the abstract according to your suggestion.
- It would be good to replace the keywords with other words than those used in the title (we will then get more phrases in search engines)
Response: We have replaced the keywords with other words according to your suggestion.
- Description needs improvement, no title for this drawing, incomprehensible descriptions on the horizontal axis (months? Stage?)
Response: We have improved description of Figure 1 and corrected the spelling error in the figure according to your suggestion. The four stages were EO treatment for 1 month, 2 months, 3 months and 4 months.
- Line 125. Lack of information what the abbreviations Rd, Rb etc. mean. Similarly in table 1.
Response: We have added the corresponding information for them according to your suggestion.
- Analysis of the oil used?
Response: The essential oil used was obtained by hydrodistillation, and the chemical composition of EOs was analyzed by GC/MS. Please check our published paper for methods and results sections.
[1] Sun W M , Ma Y N , Yin Y J , et al. Effects of Essential Oils from Zingiberaceae Plants on Root-Rot Disease of Panax notoginseng[J]. Molecules, 2018, 23(5):1021.

Reviewer 2 Report
In manuscript entitled as” Root-associated microbiomes of Panax notoginseng under the combined effect of plant development and Alpinia officinarum Hance essential oil” the authors have done nice work and in my opinion manuscript can be accepted after minor revision
My comments are following
· Please extend discussion about essential oil in introduction section.
· There are some grammatical mistake in manuscript please che3ck carefully
· What is the general molecular weight of essential weight? Please mention nin manuscript
· In manuscript Tables and Figures are not presented according to standard format please improve it
· Please improve conclusion section

Author Response
Reviewer 2:
In manuscript entitled as” Root-associated microbiomes of Panax notoginseng under the combined effect of plant development and Alpinia officinarum Hance essential oil” the authors have done nice work and in my opinion manuscript can be accepted after minor revision
Response: Thank you very much for your positive comments. According to your valuable suggestions, we have carefully revised the full text and marked it with a yellow background.
My comments are following
- Please extend discussion about essential oil in introduction section.
Response: We have extended discussion about essential oil in introduction section according to your suggestion.
- There are some grammatical mistake in manuscript please check carefully
Response: We have had our manuscript checked and corrected by using a professional language polishing service (AJE).
- What is the general molecular weight of essential weight? Please mention in manuscript
Response: EOs are a broad group of lipophilic compounds with low molecular weight. We have mentioned it in introduction section.
- In manuscript Tables and Figures are not presented according to standard format please improve it
Response: We have improved them according to your suggestion.
- Please improve conclusion section
Response: We have improved conclusion section according to your suggestion.
